# Promising Strategies of Colloidal Drug Delivery-Based Approaches in Psoriasis Management

**DOI:** 10.3390/pharmaceutics13111978

**Published:** 2021-11-22

**Authors:** Sukhbir Singh, Neelam Sharma, Tapan Behl, Bidhan Chandra Sarkar, Hasi Rani Saha, Kanika Garg, Supriya Kamari Singh, Sandeep Arora, Md. Shah Amran, Ahmed A. H. Abdellatif, Anwar L. Bilgrami, Ghulam Md Ashraf, Md. Sohanur Rahman

**Affiliations:** 1Department of Pharmacology, Chitkara College of Pharmacy, Chitkara University, Rajpura 140401, Punjab, India; sukhbir.singh@chitkara.edu.in (S.S.); neelam.mdu@gmail.com (N.S.); kanikagarg027@gmail.com (K.G.); supriyasinghh98@gmail.com (S.K.S.); sandeep.arora@chitkara.edu.in (S.A.); 2Department of Biochemistry, Primeasia University, 12- Kemal Ataturk Avenue, HBR Tower Banani C/A, Dhaka 1213, Bangladesh; bidhan.sarkar@primeasia.edu.bd (B.C.S.); hasirani85@primeasia.edu.bd (H.R.S.); 3Department of Pharmaceutical Chemistry, Faculty of Pharmacy, University of Dhaka, Shahbag, Dhaka 1000, Bangladesh; amranms@du.ac.bd; 4Department of Pharmaceutics, College of Pharmacy, Qassim University, Buraydah 51452, Saudi Arabia; a.abdellatif@qu.edu.sa; 5Department of Pharmaceutics and Industrial Pharmacy, Faculty of Pharmacy, Al-Azhar University, Assiut 71524, Egypt; 6Deanship of Scientific Research, King Abdulaziz University, Jeddah 21589, Saudi Arabia; bilgrami1956@hotmail.com; 7Department of Entomology, Rutgers University, New Brunswick, NJ 08901, USA; 8Pre-Clinical Research Unit, King Fahd Medical Research Center, King Abdulaziz University, Jeddah 21589, Saudi Arabia; ashraf.gm@gmail.com; 9Department of Medical Laboratory Technology, Faculty of Applied Medical Sciences, King Abdulaziz University, Jeddah 21589, Saudi Arabia; 10Department of Biochemistry and Molecular Biology, Trust University, Barishal, Ruiya, Nobogram Road, Barishal 8200, Bangladesh

**Keywords:** colloidal drug delivery system, dendrimers, liposome, microemulsion, nano-structured lipid carrier, psoriasis, solid lipid nanoparticles

## Abstract

Psoriasis is a chronic inflammatory autoimmune disorder that moderately affects social and interpersonal relationships. Conventional treatments for psoriasis have certain problems, such as poor drug penetration through the skin, hyper-pigmentation, and a burning sensation on normal and diseased skin. Colloidal drug delivery systems overcome the pitfalls of conventional approaches for psoriasis therapeutics and have improved patient safety parameters, compliance, and superior effectiveness. They also entail reduced toxicity. This comprehensive review’s topics include the pathogenesis of psoriasis, causes and types of psoriasis, conventional treatment alternatives for psoriasis, the need for colloidal drug delivery systems, and recent studies in colloidal drug delivery systems for the treatment of psoriasis. This review briefly describes colloidal drug delivery approaches, such as emulsion systems—i.e., multiple emulsion, microemulsion, and nano-emulsion; vesicular systems—i.e., liposomes, ethosomes, noisomes, and transferosomes; and particulate systems—i.e., solid lipid nanoparticles, solid lipid microparticles, nano-structured lipid carriers, dendrimers, nanocrystals, polymeric nanoparticles, and gold nanoparticles. The review was compiled through an extensive search of the literature through the PubMed, Google Scholar, and ScienceDirect databases. A survey of literature revealed seven formulations based upon emulsion systems, six vesicular drug delivery systems, and fourteen particulate systems reported for antipsoriatic drugs. Based on the literature studies of colloidal approaches for psoriasis management carried out in recent years, it has been concluded that colloidal pharmaceutical formulations could be investigated broadly and have a broad scope for effective management of many skin disorders in the coming decades.

## 1. Introduction

Psoriasis is a chronic inflammatory autoimmune disorder that moderately affects social and interpersonal relationships. It is rapidly built up by skin surface cells which quickly form itchy and painful red patches. An everlasting cure for psoriasis is not available; nevertheless, its effect can be decreased by quitting smoking, moisturizing, and stress management. Psoriasis is mainly proliferated by irregular keratinocyte differentiation and epidermal hyper-proliferation and is directly linked with diabetes and cardiovascular diseases [1,2,3]. It is an autoimmune acute or chronic disorder mediated by T-cells. It progresses by triggering the host’s immune system, which can be caused by keratin mediating cells multiplying and appearing on the skin’s surface. Usually, the keratinocyte cycle of development and replacement takes approximately 35–40 days, after which they shed off. However, in psoriasis, the maturation cycle takes about one week, and cells, instead of shedding off, accumulate on the skin’s surface to produce red lesions [4]. Biologics have been shown to be an excellent alternative therapy for people with moderate–severe psoriasis [5]. In the pathophysiology of psoriasis, tumor necrosis factor-α plays a crucial role. TNF-α levels are higher in psoriatics than in normal individuals, and the improvement is linked to the psoriasis area and severity index score. Psoriasis is increasingly regarded as a chronic inflammatory systemic illness mediated by multiple inflammatory cytokines, such as TNF-α, rather than a simple skin condition. TNF-α is key cytokine in psoriasis immune response activation, and also has important impacts on keratinocyte proliferation and the control of endothelium proteins required for T-cell migration [6]. Anti-TNF-α and anti-TNF-α receptor drugs are now being used to treat psoriasis [7]. 

Conventional treatments for psoriasis have certain problems, such as poor drug penetration through the skin, low aqueous solubility, poor bioavailability, asymmetric and erratic pharmacokinetic profiles, hyper-pigmentation, first-pass metabolism, and a burning sensation on normal and diseased skin. Colloidal drug delivery systems overcome the pitfalls of conventional approaches for psoriasis therapeutics. They have improved patient safety parameters, compliance, and superior effectiveness, in addition to reduced toxicity. In 2014, Marepally and colleagues incorporated two therapeutic nucleic acids, i.e., anti-STAT3 siRNA (siSTAT3) and anti-TNF-α siRNA (siTNF-α), into lipid nanoparticles using cationic amphiphilic oleyl chain-based lipids to develop fusogenic nucleic acid lipid particles (F-NALP). Topical delivery of F-NALP can transmit siSTAT3 and siTNF-α into the dermis and decrease the expression of STAT3 and TNF-α mRNAs to play synergistic role in treatment of psoriatic-like plaques [8]. Bessar et al., in 2016, developed methotrexate (MTX)-loaded gold nanoparticles functionalized with sodium 3-mercapto-1-propansulfonate (Au-3MPS@MTX conjugate). Compared to MTX alone, the conjugate was superiorly percutaneously absorbed following 24 h application on naked mouse skin, and therefore, could be explored for effective treatment of psoriasis [9]. Wan and co-researchers, in 2017, fabricated hybrid nanoparticles (FK506 NPs-NIC) composed of hyaluronic acid with cholesterol combined with nicotinamide for tacrolimus (FK506) and illustrated synergistic action on FK506 permeation through intact skin. The PASI score in a imiquimod-induced psoriasis model demonstrated that FK506 HA–Chol NP–NIC exerted an ameliorating effect on skin lesions superior to commercial FK506 ointment [10]. In 2018, Nemati et al. developed a non-toxic fusion peptide carrier, i.e., spherical nucleic acid gold nanoparticles (SNA-NCs) conjugate with siRNA, in order to enhance the penetration of siRNA into cells, and found that topical application of SNA-NCs siRNA improved psoriatic-like skin lesions via suppression of gene expression and T-cell production [11]. In 2018, a group of researchers investigated discoidal lipid nanoparticles of APTstat3 tagged with a 9-arginine cell-penetrating peptide (APTstat3-9R). APTstat3 is an inhibitor of signal transducer and activator of transcription-3 (STAT3). It was found that transcutaneous delivery of lipid nanoparticles accomplished proficient skin penetration and successfully reduced psoriatic skin inflammation without producing adverse systemic effects [12]. Ramalheiro et al., in 2020, generated encapsulated rapamycin using phytantriol-based cubosome-like liquid crystalline nanoparticles for transdermal and controlled release delivery, for psoriasis treatment, which showed a sustained drug release profile till 14 days and displayed in-vitro antiproliferative action in natural killer cells [13]. Recently, Fereig and colleagues, in 2014, developed chitosan nanoparticles with an anti-proliferative molecule, i.e., tacrolimus, which acts via T-lymphocytic cell inhibition. They reported skin deposition of 82% of the tacrolimus, which was significantly greater in comparison to pure Tacrolimus^®^ ointment, which showed about 34.0% skin retention [14]. 

Jyothi et al. also described traditional treatments along with nano-carriers and herbals used in psoriasis [15]. The current article gives brief information about conventional treatments for psoriasis and updated comprehensive information on recent advancements in various types of nanotechnology based colloidal drug delivery systems for treatment of psoriasis. The current article encompasses the colloidal drug delivery systems in several categories—viz., emulsion systems, i.e., multiple emulsion, microemulsion, and nano-emulsion; vesicular systems, i.e., liposomes, ethosomes, noisomes, and transferosomes; particulate systems, i.e., solid lipid nanoparticles, solid lipid microparticles, nano-structured lipid carriers, dendrimers, nanocrystals, polymeric nanoparticles, and gold nanoparticles; and micelle systems, i.e., polymeric micelles, reversed micelles, and mixed micelles. Moreover, the recent advancements of such colloidal drug delivery systems, i.e., emulsion systems, vesicular systems, and particulate systems, have also been summarized in tabular form. Recent studies on herbal colloidal drug delivery systems for psoriasis management have also been incorporated in the present review. For this purpose, an extensive search of the literature was conducted using PubMed, Google Scholar, and ScienceDirect databases. The keywords used in the search strategy were “psoriasis”, “liposome”, “ethosome”, “niosome”, “transferosome”, “multiple emulsions”, “microemulsion”, “nano-emulsion”, “solid lipid nanoparticles”, “solid lipid microparticles”, “nanostructured lipid carrier”, “dendrimers”, “nanocrystals”, “polymeric nanoparticle” and “gold nanoparticle” in various combinations. This should help students and research scientists to better understand future research and development in the field of colloidal drug delivery-based treatment for psoriasis.

## 2. Pathogenesis of Psoriasis

It has been commonly recognized that dendritic cells (DCs) are competent antigen-presenting cells that undertake a substantial role in several preliminary phases of the disease. The triggering of DCs in psoriasis, nevertheless, is not essentially precise. Several suggested pathways comprise the identification of antimicrobial peptides (AMPs) secreted through keratinocytes in reaction to trauma, which typically become over-expressed within psoriatic skin. LL37, S100 proteins, and β-defensins are major psoriasis-associated AMPs that are liberated through injured keratinocytes. LL-37 forms a complex with self DNA and RNA to produce LL-37-DNA and LL37-RNA complexes. The LL-37-DNA complex stimulates plasmacytoid DC (pDC) via toll-like receptor (TLR)-9, which discharges interferon-α (IFN-α) and IFN-β, leading to myeloid DC (mDC) phenotype maturation followed by Th1 and Th17 differentiation and function. Th17 cells secrete interleukins (IL)-17, IL-21, and IL-22, which activate keratinocyte proliferation in the epidermis. Complex LL-37-RNA triggers pDC via the TLR-7 pathway, which undergoes the above-described phases. Activation of mDC occurs through the TLR-8 mechanism, which migrates into draining lymph nodes that secrete tumor necrosis factor (TNF)-α, IL-23, and IL-12. These cytokines activate Th1 and Th17 differentiation and functions. Th17 cells discharge IL-17, IL-21, and IL-22, stimulating keratinocyte proliferation in the epidermis (Figure 1) [16,17,18,19]. Genetic factors also play a primitive role in the pathogenesis of psoriasis. The genetic epidemiology mainly comprises factors such as twin studies, familial aggregation, heritability, susceptibility analysis, and pedigree analysis. The tendency of family aggregation in psoriasis is highly distinctive and varies according to the population. About 31.26% of the Chinese population has reported a family history of psoriasis—67% reported having primary relatives with the disease, and 47% having secondary relatives with it. Hence, genetic predisposition has been studied, and more than 80 susceptible loci have been identified to play roles in the pathogenesis of psoriasis. The related study of gene functioning is in full swing, and several modes of next generation sequencing technology are being used to develop more accurate and sensitive genetic markers which can be targeted by biologics to impart enhanced management of disease. The genetic findings have provided hints on the pathogenesis of the disease. Disease prevention and management have been focused on by developing advanced effective biologic treatments, in response. Some of the susceptible genes that can induce psoriasis are *LCE cluster**, AIM2, LRRC7, MTHFR, MGAT5, PSORS6, SLC12A8, PSORS5, PSORS 9, and LCE1-LCE6*.

## 3. Causes and Types of Psoriasis

Stress and psoriasis are directly related to each other. Acute stress decreases one’s cortisol level, which mediates inflammation in the skin, leading to psoriasis expansion [20,21]. It has been found that psoriasis may be caused due to trauma-induced skin injuries and exposure to sunlight. Several factors, such as irritants, burns, skin tests, abrasions, and electrodesiccation, could enhance skin injuries [22,23,24,25,26]. This disease is characterized as a squamous cell disease epitomized by plaques, red, inflammatory, and silvery-white raised lesions one centimeter in diameter. The lesions on psoriasis patients that are present on the scalp are symmetrically spread. Psoriasis is normally present on the skin, scalp, elbows, sacral region, knees, tongue, or oral mucosa. On the tongue, yellow–white patches are present, which spread at a higher rate than most lesions, and they change daily. This condition is known as geographic tongue [27]. After puberty, an attack of microorganisms such as *Malassezia furfur* is responsible for seborrhea dermatitis on the scalp. The psoriatic lesions predispose all of the skin of the psoriatic patient to lesions. The part of the skin which does not involve lesions has idiopathic factors such as cell dilation and mononuclear cell infiltration [28]. Characteristic features of numerous psoriasis types, including plaques, guttate, pustular, erythrodermic, nail, and scalp psoriasis, are given in Table 1.

## 4. Conventional Treatment Alternatives for Psoriasis

Conventional psoriasis treatments consist of phototherapy, self-care, and medications that aim to confiscate psoriasis scales and prevent skin cells from growing promptly (Figure 2). Photodynamic therapy combines drugs with light therapy to destroy abnormal cells or close blood vessels. Self-care includes stress management; light therapy; ultraviolet light therapy; and applying petroleum jelly, coal tar extract, and moisturizer. Medications such as steroids work by reducing inflammation, slowing down the production of skin cells, and reducing itching. A vitamin A derivative unplugs blocked hair follicles and averts forming new blockages and decreases skin cell growth. Anti-inflammatory agents are mainly used to prevent or counteract inflammation in joints and tissues. Immunosuppressive drugs decrease the immune response. Vitamins helps with normal body functions, growth, and development. The current most effective topical treatment for psoriasis is a foam containing a steroid and a vitamin D derivative [35]. Nevertheless, all these conventional therapies have certain problems—i.e., poor drug penetration through the skin, hyper-pigmentation, and a burning sensation on normal and diseased skin in the case of topical therapy; and low aqueous solubility, poor bioavailability, and an asymmetric and erratic pharmacokinetic profile in the case of systemic therapy [1,2,3,15,26]. Systemic therapies based on traditional drugs such as cyclosporin and methotrexate, and biological therapies based on anti-tumor necrosis factor-alpha, anti-interleukin 17, and anti-interleukin 23 molecules, are used in severe cases with very good results [36,37].

## 5. The Need for Colloidal Drug Delivery Systems

Conventional treatments for psoriasis have certain problems, such as poor drug penetration through the skin, hyper-pigmentation, first-pass metabolism, and a burning sensation on normal and diseased skin. Therefore, colloidal drug delivery systems have been employed to overcome the pitfalls of conventional approaches for psoriasis treatment and improve patient safety parameters and effectiveness. Colloidal drug delivery systems have shown better drug diffusion and penetration, leading to enhanced and prolonged accumulation of drug within the skin. Moreover, drug targeting to epidermal and dermal sites can be attained, which leads to dose reduction. The colloidal formulations also minimize the systemic toxicity, burning, irritation, and necrotizing effects of certain drugs. Colloidal drug delivery has more tolerability and safety as compared to conventional treatments. The faster follicular and intercellular penetration pathway of the colloidal system is highly advantageous, as it avoids multiple administrations and improves therapeutic efficacy [1,4,15]. Figure 3 classifies various types of colloidal drug delivery systems that are highly effective for treating skin disorders. The comparative illustrations of characteristics of various colloidal carriers are depicted in Table 2 [38,39,40,41,42,43,44,45,46]. The benefits of colloidal carriers for topical drug delivery—which lead to improvements in therapeutic potential of medications for psoriasis treatment—have been summarized in Table 3 [47,48,49]. Table 4 summarizes the features of various carrier materials used in the production of colloidal systems.

## 6. Applications of Emulsion Drug Delivery Systems in Psoriasis

### 6.1. Multiple Emulsions

Emulsion drug delivery systems that can be applied for psoriasis treatments include multiple emulsion, microemulsion, and submicron emulsion systems (Figure 3). Recent advances in applications of emulsion drug delivery systems for psoriasis management are given in Table 5. Multiple emulsions are water-in-oil-in-water (w/o/w) or oil-in-water-in-oil (o/w/o) dispersed systems stabilized by hydrophilic or lipophilic surfactants. These emulsions demonstrate high potential for prolonged skin retention without enhancing its trans-dermal permeation, increased patient compliance, reduced side effects, and optimized therapeutic efficacy [38,39,40,41,42]. Laugel et al. prepared multiple topical emulsions of hydrocortisone, which showed prolonged drug release and 1.5 fold greater penetration of hydrocortisone through the epidermis compared to a simple emulsion [43,44]. 

### 6.2. Microemulsion

A microemulsion is a clear and stable system: an isotropic mixture of oil, water, and a co-surfactant with 10–100 nm globule diameters. Microemulsions are transparent or translucent, as their droplet diameter is less than a fourth of the wavelength of visible light. The microemulsion effect is governed by skin permeation, mobility, and release of drugs from the vehicle [62,63,64]. The permeation factor is influenced by alterations in the stratum cornea and numerous permeation enhancers, including short-chain fatty acids and isopropyl myristate [65]. Badawi et al. investigated salicylic microemulsions for enhanced solubility for productive topical keratolytic and antimicrobial activity [45]. Barolia et al. prepared a microemulsion of 8-methoxsalen and 5-methoxy psoralen for effective treatment of a hyper-proliferative skin disorder with improved accretion of the drug in the skin without side effects [46]. Behera et al. investigated a methoxsalen chitosan-coated microemulsion. They found decreased inflammation, prolonged drug release, and prolonged resistance of skin [59]. 

### 6.3. Nano-Emulsion

Submicron emulsions, also referred to as nano-emulsions and ultrafine emulsions, have droplets of 20–200 nm [66,67,68,69]. Drugs which have poor penetration and cause skin irritation can be delivered in nano-emulsion systems. Rajitha et al. prepared a chaulmoogra oil-loaded methotrexate nano-emulsion, which showed better skin diffusion, affluent skin retention, and 2-fold increases in antipsoriatic and anti-inflammatory properties as compared to methotrexate tablets or any marketed formulation [63]. Kaur et al. synthesized clobetasol propionate and a calcipotriol nano-emulsion to diminish inflammation as an antipsoriatic agent. They found prolonged release of the drug from the nano-emulsion gel compared to conventional medicine and any other marketed formulation [64]. 

## 7. Applications of Vesicular Drug Delivery Systems in Psoriasis

### 7.1. Liposomes

Vesicular drug delivery systems which can be beneficial for psoriasis treatments include liposomes, ethosomes, niosomes, pharmacosomes, phytosomes, and transferosomes (Figure 4). Current applications of the vesicular system for psoriasis management are given in Table 6. Liposomes are spherical-shaped small vesicles produced by cholesterol and natural, non-toxic phospholipids. These are amphiphilic molecules, as have both hydrophilic and hydrophobic ends. They are composed of bilayer components and provide rigidity and fluidity to liposome molecules (for example); unsaturated phosphatidylcholine provides permeability; and saturated phospholipids provide a rigid bilayer structure. A liposome can be a unilamellar vesicle—a single phospholipid bilayer sphere, or a multi-lamellar vesicle consisting of an onion structure [70]. Jain et al. prepared tacrolimus and curcumin-loaded synergistic liposphere gels, which showed enhanced suppression of interleukin-22, tumor necrosis factor-alpha, and interleukin-17, along with synergistic antipsoriatic activity [71]. Walunj et al. fabricated and investigated the superior efficacy of a topical gel comprising cyclosporine-loaded cationic liposomes in an imiquimod induced psoriatic plaque model [72]. 

### 7.2. Ethosomes and Niosomes

Ethanolic liposomes, referred to as ethosomes, are fabricated using ethanol as a solvent. Dubey et al. fabricated a methotrexate ethosomal gel which showed effectual antipsoriatic activity and good penetration through the skin [73]. Niosome vesicles are composed of a non-ionic surfactant. There are microscopic in size and have a lamellar structure. They can envelop a solute like liposomes, and have design flexibility. Niosomes are amphiphilic in nature, having both hydrophilic and hydrophobic characteristics. They protect the drug from the biological environment and ensure good availability (they allow little degradation). Niosomes can be used for trans-dermal and intracellular drug delivery, liver targeting, and psoriasis treatment [77,78,79]. Aggarwal et al. synthesized dithranol-loaded liposomes and niosomes, which provide localized drug delivery for psoriasis treatment along with dose reduction, thereby minimizing the burning, irritating, necrotizing, and staining effects of dithranol [75]. 

### 7.3. Transferosomes

Transferosomes are single-chain, surfactant-based, highly deface lipid vesicles that create an osmotic gradient driving force that transports the material through the skin. They provides quick invasion of the subcutaneous tissue via intracellular lipid pathway [80,81,82,83,84,85,86]. Lei et al. prepared a tacrolimus-loaded transferosome, which showed increased anti-atopic dermatitis activity compared to commercial tacrolimus ointment (Protopic^®^) liposomes-gel [75]. Gizaway et al. fabricated betamethasone di-propionate-loaded transferosomes that showed antipsoriatic activity, along with greater tolerability and better safety than the available tacrolimus formulation [76]. 

## 8. Applications of Particulate Drug Delivery Systems in Psoriasis

Particulate drug delivery systems include solid lipid nanoparticles (SLNs), solid lipid microparticles (SLMs), nanolipid carriers (NLCs), dendrimers, aquasomes, nanocrystals, polymeric nanoparticles, and gold nanoparticles (Figure 5). Current advancements of the particulate system for psoriasis therapy are described in Table 7.

### 8.1. Solid Lipid Nanoparticles (SLNs) and Solid Lipid Microparticles (SLMs)

SLNs and SLMs are colloidal, biodegradable, biocompatible, and water-based technology that avoids organic solvents. These are composed of lipids dispersed in aqueous solutions of surfactant and water. SLNs can control drug release, can improve drug stability, can confer drug targeting, have excellent biocompatibility, and can carry both lipophilic and hydrophilic drugs. SLNs are excellent topical agents because lipids are non-toxic and non-irritant, which can reduce inflammation and treat skin disorders such as psoriasis [101,102]. Ferreira et al. concluded that topical co-delivery of SLNs of methotrexate and etanercept could be a targeted approach for psoriasis management [87]. Pradhan et al. prepared fluocinolone acetonide-loaded solid lipid nanoparticles, and concluded that better penetration through the skin was achieved to treat psoriasis and hyper-proliferation keratinocytes. Furthermore, enhanced skin contact and ease of application were attained [88]. Abdel-Salam et al. prepared solid lipid nanoparticles of diflucortolone valerate and found increased anti-inflammation and skin retention compared to a normal gel [89]. 

### 8.2. Nano-Structured Lipid Carriers (NLCs)

NLCs enhance drug loading capacity and minimize drug expulsion to overcome the drawbacks of SLNs and SLMs. NLCs are composed of solid fatty acids and small amounts of liquid lipids [90,101,102,103,104]. Pradhan et al. fabricated fluocinolone acetonide loaded with nano-structured lipid carriers for effective and potential management of psoriasis [91]. Pinto et al. developed methotrexate-loaded nano-structured lipid carriers for achieving potential topical treatment of psoriasis with reduced systemic toxicity [92].

### 8.3. Dendrimers

Dendrimers are drug delivery systems consisting of highly branched molecules with a controlled, globular, reactive 3-dimensional structure with a large number of controlled peripheral functionalities. Drug molecules are either physically entrapped or covalently attached to functional groups to form drug–dendrimer conjugates [105]. Types of dendrimers include glycodendrimers, peptide dendrimers, and lysine core dendrimers. In glycodendrimers, a carbohydrate is incorporated into the dendrimer structure; i.e., the saccharine unit is present on the outer surface. In peptide dendrimers, lysine is incorporated in the core of the dendrimer structure [106]. Tripathi et al. prepared a dendrimer loaded with dithranol and found reduced burning sensation and skin irritation compared to conventional therapy [93]. Agrawal et al. explored the potential of dithranol-loaded polypropylene imine dendrimers, and found enhanced drug accumulation within the skin and enhanced targeting of epidermal and dermal sites for successful treatment of psoriasis [94,107]. 

### 8.4. Nanocrystals

Nanocrystals are nano-metric-sized crystals, i.e., 20–100 nm, which contain 100% pure drug without any conjugation with a polymer. Nanocrystals enhance solubility and dissolution by increasing the dissolution pressure, surface area, and curvature of particles, which are mainly responsible for improving the drug’s oral bioavailability [108]. Increased penetration is also achieved by nanocrystals due to the faster follicular and intercellular penetration pathway, which is highly advantageous. They have provided avoidance of multiple administration and increased residence time of the medicament [109]. Döge et al. prepared nanocrystal loaded with dexamethasone to treat lesions, and found that nanocrystal augmented the skin penetration of dexamethasone [96]. 

### 8.5. Polymeric and Gold Nanoparticles

Polymeric nanoparticles are colloidal carriers 10–1000 nm in diameter. The drug is either physically or chemically adsorbed on the surface or encapsulated inside the polymer. The silent characteristics of polymeric nanoparticles include a long shelf-life, small size, non-toxicity, and facilitating the therapeutic effect directly on the site of application. Polymeric nanoparticles can be synthesized from natural hydrophilic polymers, i.e., gelatin, albumin, alginate, chitosan, or a synthetic hydrophobic polymer—i.e., polystyrene or poly-methyl methacrylates [110]. Badilli et al. prepared poly-(d,l-lactic-co-glycolic acid) microspheres loaded with clobetasol propionate and formulated emugel, which has reduced side effects [96]. Anwer et al. developed poly (d,l-lactide-co-glycolide) nanoparticles loaded with apremilast, and found long-term retention suggesting a once-daily regimen [97]. Fratoddi et al. studied the in-vivo activity of methotrexate-loaded gold nanoparticles on an imiquimod-induced psoriasis-like mouse model, and found reductions in keratinocytes and epidermal thickness [98]. Madan et al. developed SLNs of mometasone furoate [99], and Agrawal et al. formulated NLCs of acitretin [100] to enhance the therapeutic effect of psoriasis management.

## 9. Applications of Micelle Drug Delivery Systems in Psoriasis

Micelle drug delivery systems include polymeric micelles, reverse micelles, and mixed micelles (Figure 6). Polymeric micelles are drug delivery systems with a diameter range of 10–100 nm consisting of hydrophobic cores and hydrophilic surfaces. Hydrophobic micelles are used for their stable structures, and their hydrophilic surfaces allow them to circulate in the blood for extended periods and remain unidentified by the reticuloendothelial system. Polymeric micelles can change the physicochemical structures and kinetics of drugs [111]. Polymeric micelle-incorporated drugs reach systemic circulation faster and have more excellent assembly in lymph nodes [112,113]. Lapteva et al. prepared tacrolimus-loaded polymeric micelles and concluded that tacrolimus micelles augment skin penetration by 2-fold as compared to a simple marketed formulation of tacrolimus [114]. 

## 10. Recent Advancements in Herbal Nano-Carriers for Psoriasis Treatment 

The majority of herbal drugs are insoluble; hence, they have inadequate bioavailability and augmented systemic clearance, necessitating frequent administration or else high doses. Therefore, the development of a colloidal delivery system provides several benefits to phytoconstituents, such as improved permeability, reduced toxicity, augmented pharmacological strength, fortified stability, and sustained release. Consequently, the colloidal drug delivery systems of herbal ingredients have promise for improving the effects of, and conquering the troubles linked with, herbal medicines [115,116,117,118,119,120]. Meng et al. prepared a niosome gel loaded with celastrol and concluded that celastrol loaded into noisomes leads to a two-fold increase in drug penetration into the skin [119]. Pleguezuelos-Villa et al. developed a nano-emulsion loaded with magneferan to overcome the poor aqueous solubility and low bioavailability of magneferan, which is an anti-hyperproliferative and anti-inflammatory agent. It was concluded that the nano-emulsion improved bioavailability. Therefore, it could be a better treatment for inflammatory and skin disorders [121]. Divya et al. synthesized an acitretin and aloe-emodin-loaded chitin nanogel. They found deeper penetration of aloe-emodin into the tissue, thereby producing strong anti-inflammatory action to treat psoriasis [122]. Examples of several colloidal drug delivery systems for herbal constituents used for psoriasis management are represented in Table 8.

## 11. Conclusions

Conventional therapies for psoriasis treatment have specific problems—i.e., poor drug penetration through the skin, hyper-pigmentation, and a burning sensation on normal and diseased skin in cases of topical treatment; and low aqueous solubility, poor bioavailability, and asymmetric and erratic pharmacokinetic profiles in cases of systemic therapy. Colloidal drug delivery systems for available drugs provide better drug diffusion, penetration, and targeting, leading to greater and prolonged accumulation inside the skin. Besides, these colloidal formulations can minimize the doses; systemic toxicity; and burning, irritation, and necrotizing effects of medications available for skin disorders. In the coming years, research on colloidal drug delivery system-based drug products having effective drug carriers with targeted drug release features might be extensively explored to treat several skin disease conditions.

## 12. Current and Future Developments

Current treatments for psoriasis management have some issues, such as inadequate diffusion of the medication via the skin, hyper-pigmentation, and the feeling of burning on normal and infected tissue in cases of topical therapy; and low water solubility, inadequate bioavailability, and irregular pharmacokinetics in systemic therapies. However, colloidal systems have been developed for several drugs in recent years, as described in this review. Developing a colloidal drug delivery system for available drugs could improve drug delivery, diffusion, and targeting, which might contribute to accelerated and prolonged deposition within the skin. Furthermore, such colloidal compositions can reduce the dosages, and toxic effects, burning, itching, and necrosis, caused by existing medications. Investigations on colloidal drug carrier-based pharmaceutical formulations as efficient therapeutic agents with targeted drug release characteristics could be performed widely, and could have broad applicability for the effective management of many skin disorders in the coming decades.

## Figures and Tables

**Figure 1 pharmaceutics-13-01978-f001:**
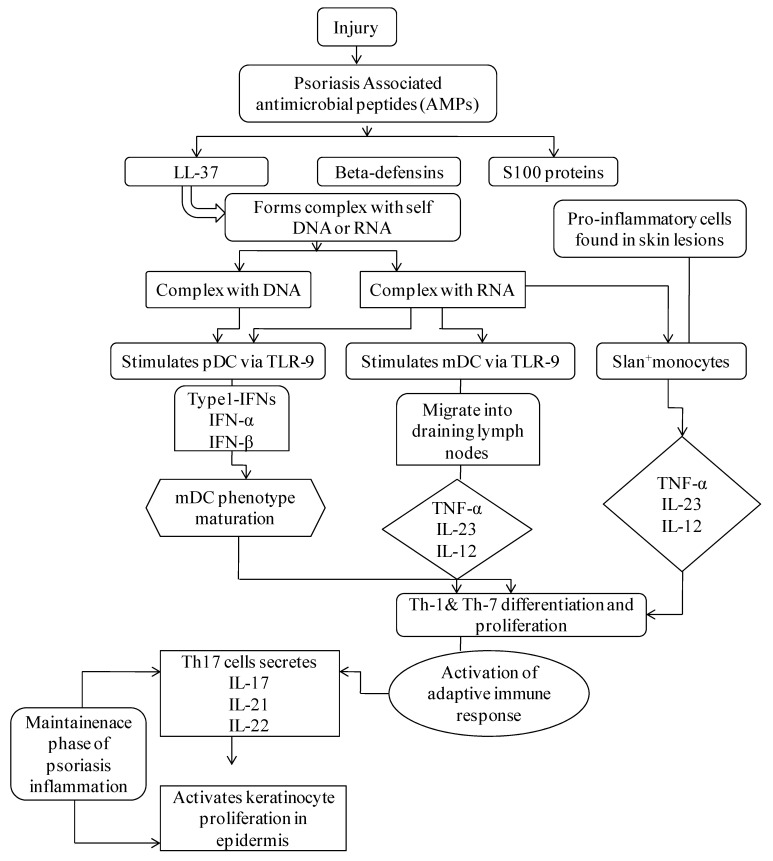
Pathogenesis of psoriasis.

**Figure 2 pharmaceutics-13-01978-f002:**
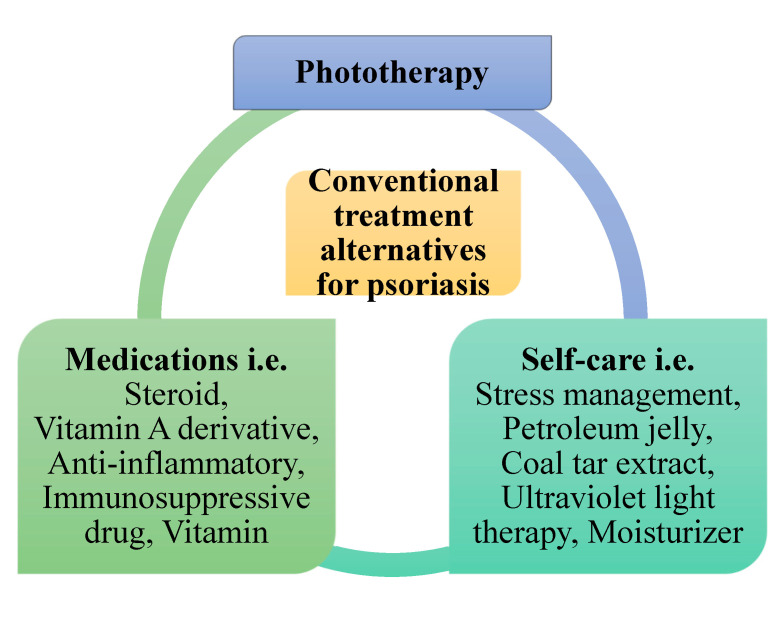
Conventional treatment alternatives for psoriasis treatment.

**Figure 3 pharmaceutics-13-01978-f003:**
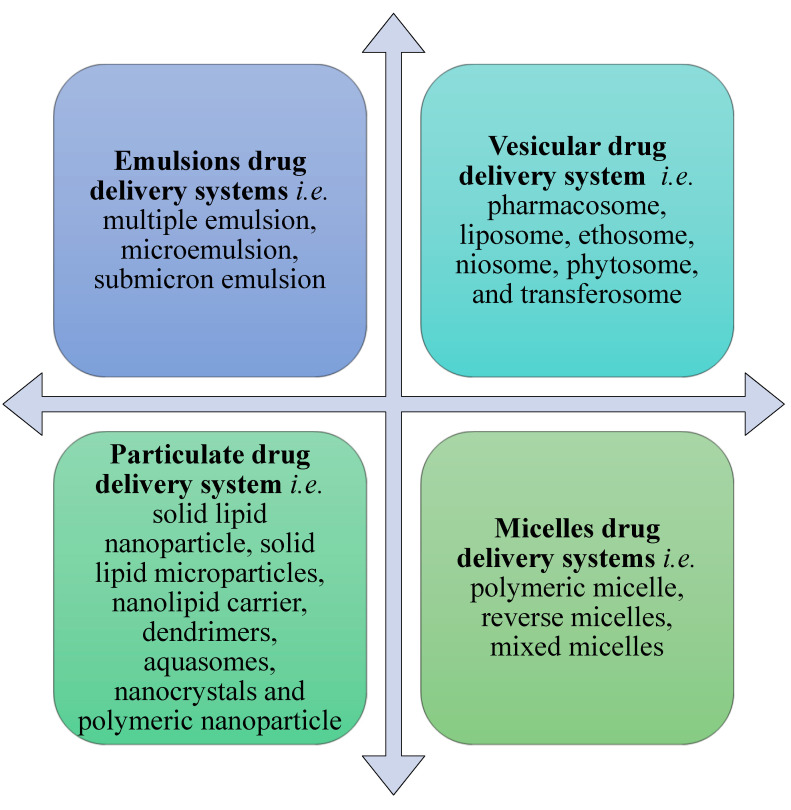
Types of colloidal drug delivery systems.

**Figure 4 pharmaceutics-13-01978-f004:**
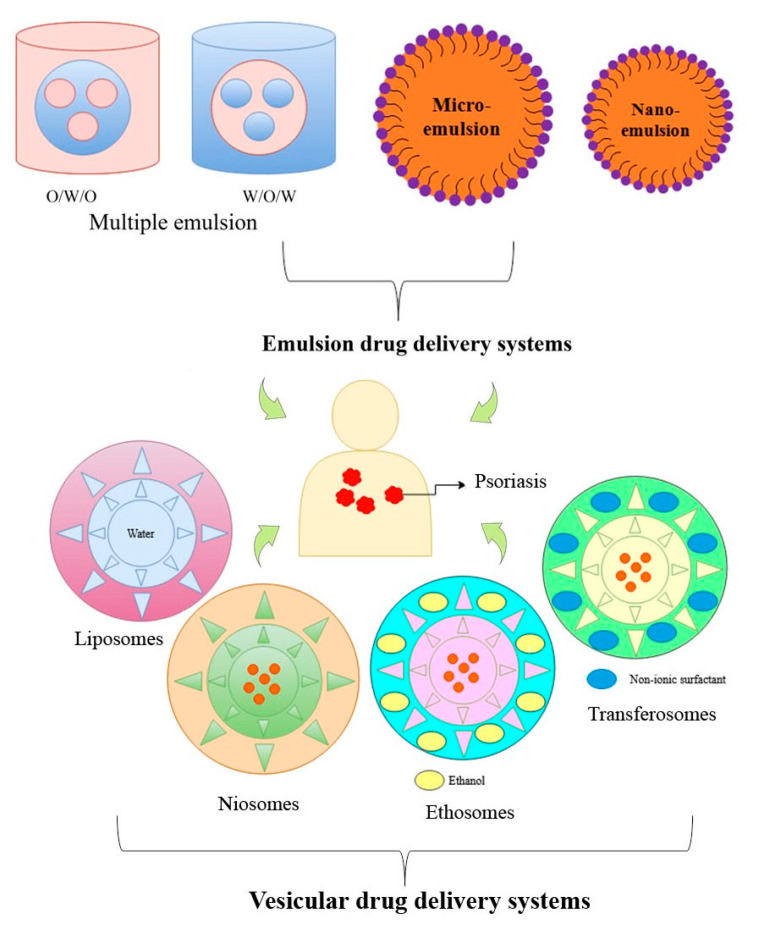
Basic structures of emulsion and vesicular drug delivery systems.

**Figure 5 pharmaceutics-13-01978-f005:**
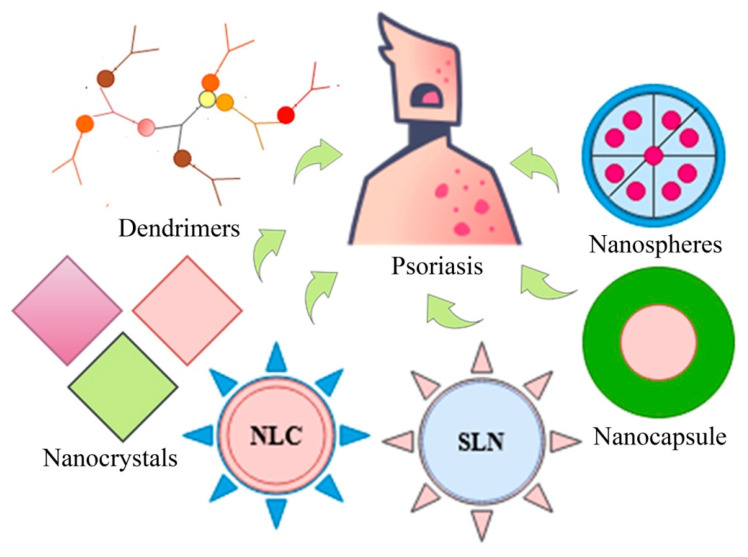
Basic structures of particulate drug delivery systems.

**Figure 6 pharmaceutics-13-01978-f006:**
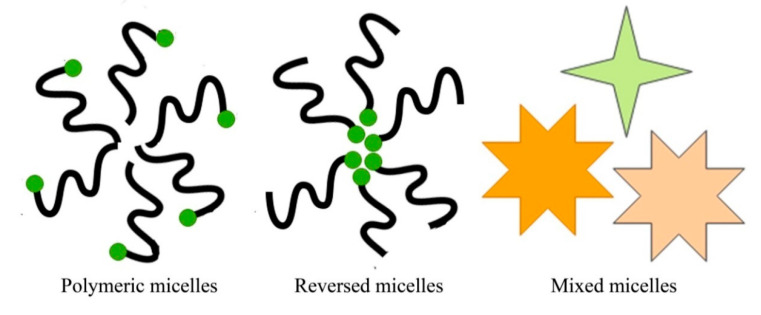
Basic structures of micelle drug delivery systems.

**Table 1 pharmaceutics-13-01978-t001:** Types of psoriasis and their characteristic features.

Psoriasis Type (Affected Area)	Characteristic Features	Ref.
Guttate psoriasis (Head and limbs)	The lesions are monomorphic. Children and adolescents are predominantly affected. The upper respiratory tract infection is occurring followed by streptococcal infection.	[15,29,30]
Plaques psoriasis/Psoriasis vulgaris (all body but typically on elbows, knees, scalp areas)	The characteristic lesions are dry, sharp and oval in shape which seems as erythematous macules and form plaque.	[15]
Pustular psoriasis (Palms and soles)	The patient suffers from red, inflamed, skin. In localized pustulosis, two distinct types of psoriasis are acrodermatitis continua of hallopeau and palmoplantar pustulosis in which pustules are formed which spread all over feet. In generalized psoriasis, pustular lesions occur in pregnancy state or due to some drugs.	[31]
Erythrodermic psoriasis (All body)	This is unstable psoriasis which is occurred by various reasons like cardiac failure, hyperthermia or deficiency of vitamins	[32,33]
Nail psoriasis (Fingernail, toe-nails)	At proximal portion of nail, small pits are forms which are characterized by orange-yellow area below nail plate which is called as oil spots.	[3]
Scalp psoriasis (Hairline)	Selection of appropriate treatment is difficult to need of application at scalp area.	[34]

**Table 2 pharmaceutics-13-01978-t002:** Distinguishing features among different types of colloidal nano-carriers.

Parameter	Phytosome	Liposome	Solid Lipid Nanoparticles	Polymeric Nanoparticle
Bond	Present	Absent	Absent	Absent
Complexity in manufacturing	Less complex	More Complex	More Complex	More Complex
Drug leakage	Less	More	More	More
Lipid drug interaction	Yes	No	No	No
Stability	More stable	Less Stable	Less Stable	More Stable
Entrapment	High	Low	Low	Low
Nature of excipients	Lipid	Lipid	Lipid	Synthetic or natural polymer
Biocompatibility	High	High	High	Low
Safety profile of solvents	ICH Class III solvents	ICH Class II and III solvents	ICH Class II and III solvents	ICH Class II and III solvents
Bioavailability	High	Moderate	Moderate	Moderate
Absorption	High	Moderate	Moderate	Moderate

**Table 3 pharmaceutics-13-01978-t003:** Advantages of colloidal drug delivery systems for topical drug delivery for psoriasis treatment.

Nanocarrier System	Advantages
Vesicular drug delivery system	
Transferosome	Large molecular weight medications are delivered to skin in a non-invasive manner.
Ethosome	Highly permeable and compliance to patient as well as safer for skin
Liposome	Amphiphilic nature, biocompatibility, and ease of surface alteration
Phytosomes	Great oral and transdermal bioavailability and therapeutic effect
Niosome	Enhances skin penetration of drugs; improved bioavailability of insufficiently absorbed drugs
Particulate drug delivery system	
Solid lipid nanoparticle	Increased efficacy, biocompatible, biodegradable
Dendrimers	Easy to prepare and alterations; better skin penetration
Polymeric nanoparticle	Used for entrapment of various class of drugs; biocompatible, biodegradable
Nanostructured lipid carriers	Reduces drug leakage during storage, increases drug payload; biocompatible
Micelles drug delivery systems	
Polymeric micelles	Thermodynamic stability, self-assembling, and skin targeting potential.

**Table 4 pharmaceutics-13-01978-t004:** Characteristics of carriers/matrices utilized in the production of colloidal drug delivery systems.

Carrier/Matrix for Nanocarrier	Properties	Ref.
Gelucires	Gelucires are polyethylene glycol glycerides made up of mono-, di-, and triglycerides, as well as mono- and diesters of polyethylene glycol.	[50]
Transcutol	Protic solvent, faint odor, colorless limpid liquid, hygroscopic having viscosity 4.1 mPa.s at 25 °C; have exceptional solubilizing capacity due to an alcohol and ether function; Used for skin penetration enhancement	[51,52]
Phosphatidylcholine	Choline is head-group of phosphatidylcholine and is attached to glycerol of fatty acids via ester-bound to backbone	[53]
Poly-lactic acid-co-glycolic acid	It is made up of glycolic acid (hydroxy acetic acid) and lactic acid (α-hydroxy propanoic acid) and due to its biocompatible and biodegradable nature, this polymer has been extensively utilized in drug delivery system with superiour loading efficiency	[54]
Cholesterol	27-carbon molecule which has amphiphilic nature and contains hydroxyl group linked with phospholipids by hydrogen bonds with the help of flexible carbohydrate linked with bulky steroid ring. During preparation of liposomes, cholesterol avoids aggregation of liposomes and enhances physical stability of membrane of liposomal vesicles and have tendency to generate stable vesicles with high drug loading capacity.	[55,56]
Chitosan	It is produced via N-deacetylation of chitin. It is biocompatible, and biodegradable polymer of natural origin, therefore considered as harmless substance for use as carrier in production of drug delivery system with high drug loading and entrapment efficiency of several drug molecules	[57]
Egg lecithin	Provides better stabilizing and encapsulation efficiency; better drug loading	[58]

**Table 5 pharmaceutics-13-01978-t005:** Recent studies in emulsion drug delivery system for psoriasis treatment.

Drug (Delivery System)	Excipients	Preparation Technique	Clinical Significance and Outcomes	Ref.
Hydrocortisone (Multiple emulsion)	Glycerol sorbitan fatty acid ester, Liquid paraffin	Oil-water-oil emulsification	Prolonged topical release of hydrocortisone in epidermis and dermis and the absorbed percentage of hydrocortisone was 1.5-fold greater from the simple emulsion compared to multiple emulsion	[43]
Cyclosporine (Micro-emulsion)	Oleic acid, Tween 80, Water, Vitamin E-TPGS, Transcutol, Propylene glycol	Emulsification	Quick skin uptake and superior skin concentrations achieved after 2 h of contact	[44]
Salicylic acid (Micro-emulsion)	Polyethylene glycol, Tween 20, Isopropyl myristate	O/W emulsification	ME with 10% SA does not show any change in storage stability after 6 months except decrease in viscosity after 1 month	[45]
8-Methoxsalen (Micro-emulsion)	Octanediol, Isopropyl myristate, Tween 80, Span 80, Water	O/W emulsification	The skin accumulation of 8-Methoxsalen was enhanced 1.5–4.5-fold	[46]
Methoxsalen (Micro-emulsion)	Egg phosphatidyl-choline, Chitosan, Ethanol, Acetic acid, Soya oil	Emulsification	Methoxsalen loaded chitosan-coated ME show controlled release of drug i.e., 18.75% lesser release than the ME with high drug retention into skin	[59]
Methotrexate (Nano- emulsion)	Tween 80, Water, Chaulmoogra oil	Self-emulsification	Showed negligible skin irritation and increased penetration into the skin	[60]
Clobitasol Propionate & Calcipotriol (Nano- emulsion)	Cremophor RH 40, Capmul MCM C8 EP, *Labrafil**^®^ M 1944 CS**,* Water	Oil in water emulsification	The nanoemulsion globules of size <100 nm also contributes to improved skin penetration, permeation and retention of drug in deep skin layers	[61]

**Table 6 pharmaceutics-13-01978-t006:** Recent studies in vesicular drug delivery system for psoriasis treatment.

Drug (Delivery System)	Excipients	Preparation Technique	Clinical Significance and Outcomes	Ref.
Tacrolimus and Curcumin (Liposphere)	Egg lecithin, Tricaprin tween 80, cremophor RH40, Isopropyl alcohol	Single emulsion solvent evaporation	Exhibited slow release of drugs and improvement in phenotypic/histopathological features of psoriatic skin	[71]
Cyclosporine (Cationic liposome)	N-(1-(2,3-dioleoyloxy) propyl)- cholesterol, chloroform, ethanol	Ethanol injection, thin film hydration, reverse phase evaporation	1.67 times rise in the levels of IL-17 on application of IMQ as compared to normal group and showed shear thinning behavior and highly effective and facilitate in psoriasis treatment	[72]
Methotrexate (Ethosome)	Soya phosphatidyl-choline, chloroform, methanol, hydro-ethanolic solution	Mechanical dispersion Cast film	Provided improved transdermal flux and reduced lag time of 0.9 h across human cadaver skin	[73]
Dithranol (Liposome, Niosome)	Phosphatidyl choline, cholesterol, span 60, chloroform	Thin-film hydration	Drug leakage study carried out at 4–8, 25 ± 2 and 37 °C for a period of two months and results revealed that leakage increased at a higher temperature and enhanced permeation with vesicles as signified through flux of dithranol	[74]
Tacrolimus (Transferosome)	Lipoid E80, Tween 80, Span 80, Dehydrated alcohol	Thin film hydration	In vitro drug release was higher in TFs-gel after 24 h in comparision to commercial ointment and from TFs-gel cumulative drug release after 12 h in vitro was 37.6%. and efficient skin target for topical delivery of tacrolimus	[75]
Betamethasone dipropionate (Transferosome)	Soya phosphatidyl-choline, Sodium deoxycholate, Tween 80, chloroform	Film hydration technique	The vesiclecs have 90.19% EE, and have great stability at 25 °C and 4 °C for 6 months and showed significant clinical improvement along with a considerable boost in safety and tolerability	[76]

**Table 7 pharmaceutics-13-01978-t007:** Recent studies in particulate drug delivery systems for psoriasis treatment.

Drug (Delivery System)	Excipients	Preparation Technique	Clinical Significance and Outcomes	Ref.
Methotrexate (SLN)	Cetyl palmitate, Polysorbate 80	Ultra-sonication	In-vitro results showed a sustained release for 8 h and enhanced skin deposition for effectual treatment of psoriasis	[87]
Fluocinolone acetonide (SLN)	Compritol 888 ATO, Soya lecithin, Poloxamer 188	Modified emulsification ultrasonication	Stability results show that SLNs were stable at 4 °C for 3 months and is a promising modality for psoriasis treatment	[88]
Diflucortolone valerate (SLN)	Geleol, Precirol ATO5, Tristearin, Compritol 888ATO, Poloxamer 407	High shear homogenization	TO produce SLNs semisolid preparation 10–20% w/w solid lipid is enough and lipid based surfactants incorporation increased entrapment efficiency and enhanced drug’s solubility	[89]
Cyclosporine and calcipotriol (SLN and NLC)	Compritol 888 ATO, Precirol, Behenic acid, Gellucire 44/14, Span 20, Cremophor RH-40, Tween 80	Hot melt homogenization	Deeper and confined drug penetration in epidermal layers for superior psoriatic management	[90]
Fluocinolone acetonide (NLC)	Compritol, Miglyol 812, Polysorbate 80	High speed homogenization	Stability findings (3 months) revealed 1.77% and 5.66% percent alteration in EE and particle size and respectively and is regarded higher potential system for psoriasis treatment	[91]
Methotrexate (NLC)	Witepsol, oleic acid, polysorbate 60, polysorbate 80	High-shear homogenization	Provided higher drug fluxes of 0.88 μg/cm ^2^ /h in comparison to free drug (0.59 μg/cm ^2^ /h) flux	[92]
Dithranol (Dendrimer)	Poly (amido) amine, ethyl cellulose, carbopol 934, Polyvinyl alcohol	Quasi-emulsion solvent diffusion	The results revealed that EE of preparation was in between 71.33% to 49.21%, particle size 28 ± 1.12 mm to 130 ± 1.01 mm and %age yield 66.28% and the formulation produced prolonged efficacy without causing skin toxicities	[93]
Dithranol (Dendrimer)	Ethylenediamine	Divergent method	Primary irritation index of DIT–PPI was revealed to be 1 that means DIT-PPU causes less irritation and have high drug penetration in controlled manner	[94]
Dexa-methasone (Nano-crystal)	Polyvinyl alcohol, Sodium lauryl sulphate	Wet bead milling	Superior drug penetration and distribution within skin with reduced dose	[95]
Clobetasol propionate (Polymeric microsphere)	Poly (d,l-lactide co-glycolide), polyvinyl alcohol	Solvent evaporation	F8-coded preparation fabricated using PLGA 50:50 at 1:5 drug/polymer ratio and homogenised for 1 min at rpm 8000 was considered as the best preparation and having superior drug efficacy in topical applications	[96]
Apremilast (Polymeric nano-particle)	Poly (d,l-lactide co-glycolide), Polyvinyl alcohol	Single emulsion solvent evaporation	2.25 folds increment in bio-availability of F3 nanoparticles than normal APM suspension and enhancement in half-life and mean residence time leads to long-term retention of nano-particles to provide once-daily regimen	[97]
Methotrexate (Gold nano-particle)	3-mercapto-1-propansulfonate, Diethylaminoethanethiol hydrochloride, tetrachloroauric (III) acid, sodium borohydride	Bioconjugation and functionalization	Induced a diminution of keratinocytes hyper-proliferation, epidermal thickness as well as inflammatory infiltrate *in* imiquimod-induced psoriasis like mice model	[98]
Mometasone furoate (SLN)	Glycerol mono-stearate, Tefose-63, Tween-80	Solvent injection method	2.67 times more skin deposition as compared to marketed cream and 20 times more in comparsion to plain drug loaded gel and is promising for topical delivery of corticosteroid	[99]
Acitretin (NLC)	Precirol ATO5, oleic acid, tween 80, tetrahydrofuran	Solvent diffusion method	Increase in Acitretin deposition was found in cadaver skin from ActNLC gel (81.38 ± 1.23%) than to Act plain gel (47.28 ± 1.02%) and enhancement in therapeutic effect for psoriasis and decrease in local side effects	[100]

**Table 8 pharmaceutics-13-01978-t008:** Recent works in herbal colloidal drug delivery systems for psoriasis management.

Herbal Constituent (Delivery System)	Excipients	Preparation Technique	Clinical Significance and Outcomes	Ref.
Celastrol isolated from Tripterygium regelii (Niosome)	Cholesterol, carbopol 934, span 20, span 60	Thin film hydration	The developed nanoparticle has particle size of 147 nm and yield of up to 90% and increased water solubility and permeation of celastrol into skin which enhanced its anti-psoriasis activity in mice	[123]
Mangiferin isolated from leaves/bark of Mangifera indica (Nano-emulsion)	Lipoid^®^ S75, hylouronic acid, Polysorbate 80	Ultra-sonication	Nanoemulsions having mangiferin significantly reduce oedema ∼20-fold higher than empty nanoemulsions and reduce leucocyte infiltration and showed an anti-inflammatory activity	[124]
Acitretin and aloe-emodin Aloe-emodin isolated from plant of genus Aloe (Polymeric nanoparticle)	Chitin	Centrifugation	Revealed greater skin permeation and drug retention in deeper layer of skin with and improve compatibility	[125]
Tea tree oil isolated from leaves of Melaleuca alternifolia (Micro-emulsion)	Tween 80	Emulsification	Showed superior drug solubilization and bioavailability for topical applications of anti-psoriatic active moieties and bio-actives	[126]
Curcumin (Nano-hydrogel)	Curcumin, choline-calix[4]arene amphiphile	Supra-molecular nano-hydrogel	Exhibited no significant toxicity and showed effective anti-psoriatic activity in an IMQ-induced psoriasis mouse via decreased pro-inflammation	[127]

## Data Availability

Not applicable.

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
