# Peer review of "Promising Strategies of Colloidal Drug Delivery-Based Approaches in Psoriasis Management"

_pharmaceutics, 2021, doi:10.3390/pharmaceutics13111978_

Round 1
Reviewer 1 Report
The manuscript focuses on colloidal drug delivery of psoriasis treatment. There were several major issues with the manuscript:
(1) The authors made the review manuscript toward the review of all possible colloidal drug delivery systems, rather than the review of the promising therapeutic methods for psoriasis. If all methods were required to be provided, please give a table comparison on various colloidal drug delivery systems on the effects of therapeutic effect of psoriasis to shorten the content discussed.
(2) Colloidal drug delivery systems require carriers. The authors should spend a quality amount of the content to address the physicochemical properties of the carrier/matrix as well as how the loadings and encapsulation efficiencies were compared among various systems.
(3) Table 3 - 6 only listed qualitative data. Please revise them to include quantitative data. The authors should know that it will be a lengthy manuscript by doing so. Therefore, please consider to shorten the contents and acquire a focal point on the review.
Thank you,
Reviewer
Author Response
I have revised the manuscript according to your comments and enclosed the response to the cover letter.
Reviewer 2 Report
The manuscript has been improved once resubmitted and it is suitable for pubblication
Author Response

(The authors gave the same response as above.)

Reviewer 3 Report
A review exploring drug delivery approaches for topical medications in psoriasis. Although the article is well structured considering only the part of the carriers, the article has different problems and different errors from a dermatological point of view( so basically in the part where you describe psoriasis and its current therapies).
Major Revisions are required before possibly considering this article for publication.
One of the most important problems is the fact that photodynamic therapy(PDT) is not a treatment used to manage psoriasis;PDT uses light at 630 nm to activate a substance, usually 5 aminolevulinic acid, in protoporphyrin 9; this treatment is usually used to treat precancerosis and superficial cutaneous skin cancers. The therapy that you are referring to is probably phototherapy, which uses narrowband ultraviolet B light to treat different diseases, such as psoriasis( you also referred it as a possible treatment).
Page 6 line 127-129 please delete
line 112-114 "Furthermore, smoking, alcoholism, and various drugs such as lithium therapy, cardioselective beta-1 blocker, and non-cardio selective beta-blocker are also responsible for psoriasis" this is not true, an association is present but they are not responsible for psoriasis; please correct or delete.
line 115 "psoriasis may be persuaded"????
Psoriasis is a squamous condition; papules are very rarely present. Please modify.
line 138 "Medications such as steroids alter or stimulate hormone effects and further diminish inflammation." I do not understand what you want to say...please reassess.
When you talk about current psoriasis treatment you must always specify that you are talking about topical treatments. There are a lot of very effective systemic treatments that you should at least mention in the introduction part ( you mention them very out of the contest in chapter 11......this must be assessed.)
Table 1 Pustular psoriasis is not well described in the references you cited; you should add a proper reference, such as: doi: 10.1111/dth.13170.
In chapter 4, conventional treatment for psoriasis you should add: "the current most effective topical treatment for psoriasis is a combination foam of a steroid and a vitamin D derivative" and add a proper citation, such as: doi: 10.1111/dth.13185.
at the end of chapter 4 you should at least mention systemic therapies (although your article focuses on topical ones): "Systemic therapies based on traditional drugs such as cyclosporin and methotrexate, and biological therapies based on anti-Tumour Necrosis factor-alpha, anti-Interleukin 17 and anti-Interleukin 23 molecules are used in severe cases with very good results" and add proper citations, such as: doi: 10.3390/healthcare9050543. and doi: 10.1111/dth.14504.
Thank You
Author Response

(The authors gave the same response as above.)

Round 2
Reviewer 1 Report
The authors have addressed my recommendations. Please proofread the manuscript again.
Reviewer 3 Report
The article improved a lot after revisions...there are still some typos and minor errors, but I think the paper is now publishable.
This manuscript is a resubmission of an earlier submission. The following is a list of the peer review reports and author responses from that submission.
Round 1
Reviewer 1 Report
The manuscript in the present form is not suitable for the pubblication in Pharmaceutics journal. There is a lack of critical comparisions among the colloidal drug delivery systems presented as effective approaches for the treatment of psoriasis. It appears as a mere list of colloidal drug delivery systems, reporting in each paragraph few examples of experimental works without any discussion and deep considerations. The cited works are not exhaustive and span over a long period, without a particular attention on the recent literature. The figures also have not a sufficient scientific clearness and soundness.
Reviewer 2 Report
The review manuscript was well-written and well-organized. Please check grammar errors and typos to further improve the manuscript.
Reviewer 3 Report
This is a manuscript on colloidal drug delivery systems specifically for psoriasis treatment. Unfortunately, the authors did not cover the genetics of psoriasis which is one of the major factors. It is not necessary that psoriasis would always be linked to some injury. At present, majority of the treatments for psoriasis is based on biologics that have minimal emphasis in this article. In fact, there are not too many colloidal systems for biologics anyway.
Colloidal drug delivery part of this review article could be well-known facts for the readership of this journal.